# Factors Related to Plasma Homocysteine Concentration in Young Adults: A Retrospective Study Based on Checkup Populations

**DOI:** 10.3390/jcm12041656

**Published:** 2023-02-19

**Authors:** Zhihua Li, Jing Zhao, Chengbei Hou, Fei Sun, Jing Dong, Yansu Guo, Xi Chu

**Affiliations:** 1Information Center, Xuanwu Hospital, Capital Medical University, No. 45 Changchun Street, Xicheng District, Beijing 100053, China; 2Health Management Department, Xuanwu Hospital, Capital Medical University, No. 45 Changchun Street, Xicheng District, Beijing 100053, China; 3Department of Evidence-Based Medicine, Xuanwu Hospital, Capital Medical University, No. 45 Changchun Street, Xicheng District, Beijing 100053, China; 4Beijing Geriatric Healthcare Center, Xuanwu Hospital, Capital Medical University, No. 45 Changchun Street, Xicheng District, Beijing 100053, China

**Keywords:** homocysteine, hyperhomocysteinemia, young adults, correlation analysis, BMI

## Abstract

The distribution profile of plasma homocysteine (Hcy) in young adults and its related factors are not well understood. We performed a generalized estimating equations (GEE) analysis for plasma-Hcy-correlated factors in 2436 young adults, aged 20–39 years, from a health checkup population. We observed that the mean Hcy concentration in males (16.7 ± 10.3 μmol/L) was significantly higher than that in females (10.3 ± 4.0 μmol/L), and hyperhomocysteinemia (HHcy) prevalence in males was 5.37 times than that in females (33.3% vs. 6.2%). A GEE analysis stratified by sex indicated that age (B = −0.398, *p* < 0.001) and LDL-C (B = −1.602, *p* = 0.043) were negatively correlated, while BMI (B = 0.400, *p* = 0.042) was positively correlated, with the Hcy level in young males. ALT (B = −0.021, *p* = 0.033), LDL-C (B = −1.198, *p* < 0.001) and Glu (B = −0.446, *p* = 0.006) were negatively correlated, while AST (B = 0.022, *p* = 0.048), CREA (B = 0.035, *p* < 0.001), UA (B = 0.004, *p* = 0.003) and TG (B = 1.042, *p* < 0.001) were positively correlated, with the Hcy level in young females. These results suggest that young males have a significantly higher plasma Hcy level and HHcy prevalence than young females; therefore, more attention should be paid to the reason for and effect of the higher HHcy prevalence in young males.

## 1. Introduction

Homocysteine (Hcy) is a sulfur-containing amino acid and a metabolic intermediate in methionine metabolism, crucial for regulating methionine availability, protein homeostasis and DNA methylation reactions [1]. Hcy metabolism disruption is relevant to a number of pathological conditions, including endothelial dysfunction and atherosclerotic vascular lesions [2]. Elevation in Hcy can be caused by a decline in renal function, the insufficiency or impaired function of B vitamins, the dysregulation of the methionine cycle, etc. [3]. The blood level of Hcy has been reported to increase with age [4,5]. However, in our previous study, only in all-age females was a positive correlation between the plasma Hcy concentration and age found; in addition, we unexpectedly noticed that young male adults < 30 years old in the checkup population showed similarly high Hcy levels to the older groups. Understanding whether this phenomenon is caused by enrollment bias deserves further study.

Meanwhile, it was recently shown that in young adults (≤35 years of age), hyperhomocysteinemia (HHcy) was significantly associated with the presence of acute coronary syndrome and the severity of coronary artery stenosis [6]. HHcy is an independent risk factor for coronary artery disease (CAD) in young Chinese patients [7], and the combination of HHcy and smoking is a significant risk factor for the severity of CAD [8]. A case-control study also indicated a strong association between plasma Hcy and a first myocardial infarction in young patients (≤45 years of age) [9]. In addition, HHcy was reported to be one of the risk factors in young intracranial large-artery atherosclerotic stroke patients [10]. Although a number of studies on the risk factors of HHcy have been reported in middle-aged or older populations [11], the distribution profile of plasma Hcy levels and the risk factors of HHcy among young adults have not drawn much attention.

The aim of the present study was to determine the distribution profile and related factors of the plasma Hcy level in young adults (<40 years of age) using longitudinal retrospective health checkup data from 2012 to 2020, in which the participants underwent at least two repeated checkups during this period.

## 2. Materials and Methods

### 2.1. Study Sample

All analyses in this study were performed on physical checkup data from the period of January 2012 to December 2020 collected by Xuanwu Hospital, Capital Medical University, China. Subjects who were 20–39 years old, underwent ≥2 yearly routine medical checkups during this period and completed the laboratory measurements, including plasma Hcy, alanine aminotransferase (ALT), aspartate aminotransferase (AST), creatinine (CREA), uric acid (UA), triglyceride (TG), total cholesterol (TC), low-density lipoprotein cholesterol (LDL-C), high-density lipoprotein cholesterol (HDL-C), and glucose (Glu), were recruited. In addition, information on body mass index (BMI) and the waist-to-hip ratio (WHR) was available. Subjects with known chronic renal failure and cancer were excluded. If ≥2 checkups were undertaken for a subject in a year, only the first checkup data with all the indexes of interest available were analyzed. 

### 2.2. Measurements

Data were collected by trained staff from the Health Management Department, Xuanwu Hospital. Fasting (≥8 h) blood samples were collected for biochemical analysis on a Hitachi 7600 automated biochemical analyzer (Hitachi, Tokyo, Japan). Plasma Hcy levels were measured using an enzymatic cycling method at Xuanwu Hospital, Capital Medical University. The diagnostic criterion for HHcy was defined as >15 μmol/L [12].

### 2.3. Statistical Analysis

Descriptive statistics are expressed as means ± standard deviations (SDs) for continuous variables, and as frequencies and percentages for categorical variables. Group differences between males and females were tested using an independent t-test. The chi-square test was used to compare the percentages of HHcy in males and females. ANOVA and post hoc least significant difference (LSD) tests were used to determine the differences between groups. Pearson correlation analysis was performed to present the association of the Hcy level with each variable. The trends of Hcy in both genders were assessed using generalized estimating equation models with a linear link and normal distribution in order to address the correlation of repeated measurements within the same participants in waves. The models were adjusted for BMI, WHR, ALT, AST, CREA, UA, TG, TC, LDL-C, HDL-C and Glu. Different working correlation matrix structures were compared with the use of the corrected quasi likelihood under the independence model criterion. The chosen model was tested with an unstructured matrix. All the statistical tests were two-tailed, and *p* < 0.05 was considered to be statistically significant. All analyses were performed using SPSS software version 23.0.

### 2.4. Ethical Considerations

This study was approved by the Ethics Committee of Xuanwu Hospital, Capital Medical University, China. It was a retrospective observational study, and was exempt from the written informed consent requirement. The procedures complied with the World Medical Association’s Declaration of Helsinki regarding the ethical conduct of research involving human subjects.

## 3. Results

### 3.1. Characteristics of the Subjects

A total of 2436 subjects from the health checkup data of the Health Management Department of Xuanwu Hospital were included in this study, and the mean age was 31.5 ± 4.2 years. Of these subjects, 1245 (51.11 %) were males (31.7 ± 4.0 years) and 1191 (48.89 %) were females (31.3 ± 4.4 years). Each subject underwent 2–8 yearly checkups from 2012 to 2020. Altogether, 6449 checkups of 2436 participants were analyzed. The mean Hcy level of all the subjects was 13.6 ± 8.5 μmol/L (16.7 ± 10.3 μmol/L for males and 10.3 ± 4.0 μmol/L for females), and the overall prevalence of HHcy was 20% (33.3% for males and 6.2% for females). The Hcy level and HHcy prevalence in males were significantly higher than those of females (*p* < 0.001) (Table 1).

### 3.2. Distribution Profile of Hcy Levels and HHcy Prevalence in Different Age Groups of Young Adults

All of the subjects were assigned to four age groups: 20–24 years (20–24 Y), 25–29 years (25–29 Y), 30–34 years (30–34 Y) and 35–39 years (35–39 Y), with 125, 674, 934 and 703 participants, respectively (Table 1). The mean Hcy levels in these age groups were 18.1, 18.9, 16.2 and 15.2 μmol/L in males, and 11.0, 11.0, 10.2 and 9.6 μmol/L in females, respectively. The Hcy levels showed an overall descending trend with increasing age between 20 and 39 years in both males and females. Consistent with the Hcy levels, HHcy prevalence declined with age in both genders, with percentages of 45.5%, 41.6%, 31.2%, and 26.9% in males, and 8.6%, 8.1%, 6.6% and 3.2% in females, in the 20–24, 25–29, 30–34 and 35–39 age groups, respectively. In each age group, the Hcy level and HHcy prevalence of males were both significantly higher than those of females (*p* < 0.001) (Table 1). ANOVA analysis indicated that the 25–29 age group had a significantly higher Hcy level than both the 30–34 and 35–39 age groups in both males (*p* < 0.001) and females (*p* = 0.008; *p* < 0.001). Meanwhile, in females, the 20–24, 25–29 and 30–34 age groups showed a significantly higher Hcy level than the 35–39 age group (*p* = 0.007; *p* < 0.001; *p* = 0.036) (Figure 1). Correlation analyses between the Hcy level and each variable in each age group are shown in Appendix A.

### 3.3. Generalized Estimating Equations Analysis of Longitudinal Repeated Data for Hcy-Correlated Factors in Young Adults

For the longitudinal analysis of the 6449 checkups of 2436 participants (each subject underwent ≥2 repeated yearly checkups between 2012 and 2020), the generalized estimating equations (GEE) approach was used to investigate the related factors of the plasma Hcy level in young adults. In total, 3295 checkups of 1245 males and 3154 checkups of 1191 females were analyzed. In males, GEE analysis indicated that age (B = −0.398; *p* < 0.001) and LDL-C (B = −1.602; *p* = 0.043) was negatively correlated with the Hcy level, while BMI (B = 0.400; *p* = 0.042) was positively correlated with the Hcy level (Table 2). However, in females, ALT (B= −0.021; *p* = 0.033), LDL-C (B= −1.198; *p* < 0.001) and Glu (B= −0.446; *p* = 0.006) were negatively associated with the Hcy level, while AST (B = 0.022; *p* = 0.048), CREA (B = 0.035; *p* < 0.001), UA (B = 0.004; *p* = 0.003) and TG (B = 1.042; *p* < 0.001) were positively associated with the Hcy level. Age was not correlated with Hcy levels in young female adults (Table 3).

## 4. Discussion

To our knowledge, this is the first study that investigates the distribution profiles and related factors of Hcy levels in young adults <40 years of age from regular checkup data using longitudinal repeated data. In this study, we found that, in the young adults, males had a significantly higher Hcy level and HHcy prevalence than females. In addition, in young males, age and LDL-C were negatively correlated with the Hcy level, and BMI was a risk factor for a high Hcy concentration. The results widen our knowledge of the distribution profile of plasma Hcy concentrations in young adults, suggesting that more attention needs to be paid to young males with elevated Hcy levels.

Several studies investigated the Hcy concentration in the young with cardiocerebrovascular diseases. Wu et al. [7] showed that young patients with coronary artery disease (CAD) (aged ≤55 years) had a higher Hcy concentration (16.2 μmol/L) than the normal control group (12.1 μmol/L). Similarly, a higher Hcy level (20.79 μmol/L) in young ischemic stroke patients (aged ≤45 years) and a much lower Hcy level (11.10 μmol/L) in age-and gender-matched healthy controls were reported by Rudreshkumar et al. [13]. In our study, males exhibited a much higher Hcy concentration (16.7 μmol/L), which was 1.6 times that of females (10.3 μmol/L). Meanwhile, the prevalence of HHcy among young males was 5.4 times that of young females (33.3% vs. 6.2%). Therefore, more attention should be paid to the elevated Hcy levels in young male adults. Studies on the reason for and effect of elevated Hcy levels would provide more profound indications for preventing HHcy-related pathological conditions.

Age is commonly recognized as a related factor of Hcy concentrations and impacts on the Hcy metabolism dynamics [14]. In our previous study, we found that every increase of 1 year in age increased the Hcy concentration by 0.051 μmol/L in the female checkup population (aged 16–102 years). Unexpectedly, no association between age and Hcy levels was observed in the all-age male checkup population, and a particularly high Hcy concentration in the 16–29 years age group was found in males [4]. Therefore, in the present study, special attention was paid to young adults <40 years, and a narrower age bracket was used. In accordance with our previous results [4], males showed a much higher Hcy level (1.6–1.7 times) and HHcy prevalence (4.7–8.4 times) than females in all of the included age groups in this study. Moreover, age was found to be negatively related to the Hcy concentration in young males (<40 years of age). The investigation of whether the elevated Hcy level in young males is physiologic or pathologic deserves further study.

Obesity has been reported to affect Hcy levels, and obese patients have been found to have a significantly elevated Hcy concentration [15]. The association between lipid profiles and Hcy levels has been extensively studied, but the conclusions have been inconsistent. In a study of 4012 Chinese people aged 30–92 years by Niu et al. [16], it was found that HHcy was independently associated with high levels of LDL-C, TG and TC in both genders, and low levels of HDL-C in females. Our previous study of 7838 males and 7073 females from a regular checkup population aged 16–102 years found that high levels of LDL-C and HDL-C and a low level of TC in both genders, as well as high levels of TG in females, were risk factors for high Hcy concentrations [4]. In this study, high levels of BMI and low levels of LDL-C were found to be risk factors for high Hcy concentrations in young males. Low levels of LDL-C and high levels of TG were associated with elevated Hcy concentrations in young females. Taken together, our studies indicate that low LDL-C levels are associated with higher Hcy concentrations regardless of gender in young adults, and that a high TG level is an independent risk factor for high Hcy concentrations in females at all ages. To summarize, the associations between lipids and Hcy concentrations are distinct according to the gender difference and age range.

It is commonly known that Hcy levels are correlated with glomerular filtration [17]. Accordingly, in this study, CREA and UA were found to be independently associated with Hcy concentration in young females. Moreover, Glu and ALT were observed to be negatively related, while AST was observed to be positively related, to the Hcy level in young females; this trend is in accordance with our previous all-age checkup population study [4]. However, no significant association between Hcy concentrations and CREA, EA, ALT, AST and Glu existed in young male adults.

### Study Strength and Limitation

There are several strengths in this study. First, the risk factors for high Hcy concentrations were focused on young adults, and each subject underwent more than two repeated yearly checkups within an 8-year period. Second, combined with our previous study [4], gender- and age-specific risk factors for high plasma Hcy levels were identified in a regular checkup population. The limitations of our study include the lack of lifestyle behaviors and nutritional conditions. Furthermore, information about the genetic profiles of the participants was unavailable. Finally, our study is limited by the fact that our analysis did not take into account the status of cardiac and cerebral vessels.

## 5. Conclusions

In summary, this study presents the distribution profiles of the Hcy concentration and HHcy prevalence in young adults, and also identifies gender-specific risk factors for high Hcy levels in the young (aged <40 years). Young males had a significantly higher plasma Hcy level and HHcy prevalence than young females. Age and LDL-C were negatively related, while BMI was positively related, to the Hcy level in young males. Prospective studies need to be performed to elucidate the reason for and effects of elevated Hcy concentrations in young males.

It is important to note that gender and age may have an indispensable influence on the plasma Hcy concentration. The normal Hcy concentration range may vary according to different gender and age brackets; therefore, the reference values for Hcy according to gender and age should be revised.

## Figures and Tables

**Figure 1 jcm-12-01656-f001:**
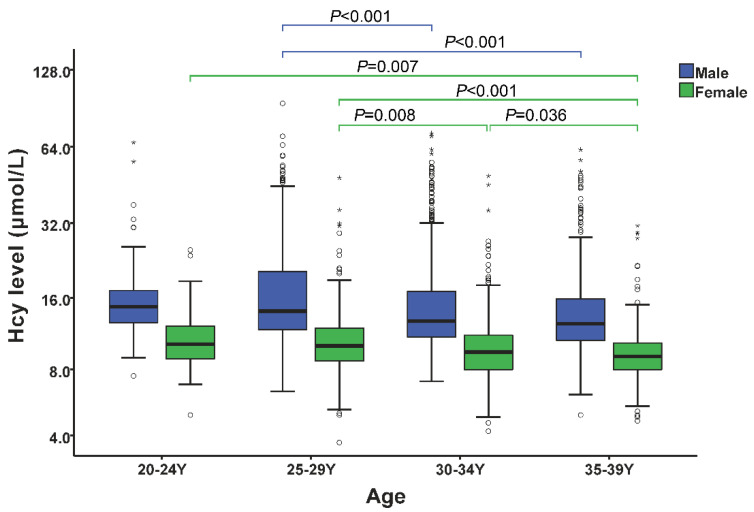
Comparison of plasma Hcy levels among different age groups in males and females. The boxplot shows the distribution of the plasma Hcy concentrations of all the subjects, as well as the medians and quartiles. ANOVA and post hoc least significant difference (LSD) tests were used to compare the differences between groups. Hcy, homocysteine; Y, years. Open circles and stars represent outliers and extreme outliers.

**Table 1 jcm-12-01656-t001:** Distribution profiles of Hcy levels and HHcy prevalence in young adults less than 40 years of age.

Study Group		Overall	Male	Female	*p* Value	Difference	95%CI
Sum (n)		2436	1245	1191			
	Age	31.5 ± 4.2	31.7 ± 4.0	31.3 ± 4.4			
	Hcy (μmol/L)	13.6 ± 8.5	16.7 ± 10.3	10.3 ± 4.0	<0.001	6.4	5.8–7.0
	HHcy (%)	488 (20%)	414 (33.3%)	74 (6.2%)	<0.001		
20–24 Y (n)		125	55	70			
	Hcy (μmol/L)	14.1 ± 8.8	18.1 ± 11.7	11.0 ± 3.4	<0.001	7.1	3.8–10.3
	HHcy (%)	31 (24.8%)	25 (45.5%)	6 (8.6%)	<0.001		
25–29 Y (n)		674	317	357			
	Hcy (μmol/L)	14.7 ± 9.8	18.9 ± 12.2	11.0 ± 4.4	<0.001	7.9	6.5–9.3
	HHcy (%)	161 (23.9%)	132 (41.6%)	29(8.1%)	<0.001		
30–34 Y (n)		934	509	425			
	Hcy (μmol/L)	13.5 ± 8.3	16.2 ± 9.7	10.2 ± 4.3	<0.001	6.0	5.0–6.9
	HHcy (%)	187 (20.0%)	159 (31.2%)	28(6.6%)	<0.001		
35–39 Y (n)		703	364	339			
	Hcy (μmol/L)	12.5 ± 7.1	15.2 ± 8.6	9.6 ± 3.2	<0.001	5.6	4.7–6.6
	HHcy (%)	109 (15.5%)	98 (26.9%)	11(3.2%)	<0.001		

Notes: Values are expressed as the mean ± SD. Hcy, homocysteine; HHcy, hyperhomocysteinemia; Y, years; CI, confidence interval.

**Table 2 jcm-12-01656-t002:** Longitudinal analysis of Hcy-correlated factors in young males using generalized estimating equations.

Variable	B	95% Confidence Interval		*p* Value
		Lower	Upper	
Age	−0.398	−0.591	−0.205	<0.001 **
BMI	0.400	0.014	0.786	0.042 *
WHR	−8.095	−18.719	2.528	0.135
ALT	−0.060	−0.145	0.025	0.164
AST	0.082	−0.055	0.220	0.242
CREA	−0.023	−0.053	0.008	0.152
UA	0.003	−0.004	0.009	0.440
TG	0.016	−1.324	1.355	0.982
TC	0.079	−0.221	0.379	0.606
LDL-C	−1.602	−3.154	−0.050	0.043 *
HDL-C	1.323	−0.276	2.923	0.105
Glu	−0.125	−0.551	0.301	0.565

Notes: BMI, body mass index; WHR, waist-to-hip ratio; ALT, alanine aminotransferase; AST, aspartate aminotransferase; CREA, creatinine; UA, uric acid; TG, triglyceride; TC, total cholesterol; LDL-C, low-density lipoprotein cholesterol; HDL-C, high-density lipoprotein cholesterol; Glu, glucose. *, *p* < 0.05; **, *p* < 0.01.

**Table 3 jcm-12-01656-t003:** Longitudinal analysis of Hcy correlated factors in young females using generalized estimating equations.

Variable	B	95% Confidence Interval		*p* Value
		Lower	Upper	
Age	−0.043	−0.089	0.002	0.059
BMI	0.060	−0.010	0.130	0.096
WHR	−0.802	−4.680	3.075	0.685
ALT	−0.021	−0.040	−0.002	0.033 *
AST	0.022	0.000	0.045	0.048 *
CREA	0.035	0.016	0.054	<0.001 **
UA	0.004	0.001	0.006	0.003 **
TG	1.042	0.565	1.518	<0.001 **
TC	−0.104	−0.462	0.254	0.568
LDL-C	−1.198	−1.724	−0.672	<0.001 **
HDL-C	−0.062	−0.616	0.491	0.825
Glu	−0.446	−0.766	−0.126	0.006 **

Notes: BMI, body mass index; WHR, waist-to-hip ratio; ALT, alanine aminotransferase; AST, aspartate aminotransferase; CREA, creatinine; UA, uric acid; TG, triglyceride; TC, total cholesterol; LDL-C, low-density lipoprotein cholesterol; HDL-C, high-density lipoprotein cholesterol; Glu, glucose. *, *p* < 0.05; **, *p* < 0.01.

## Data Availability

The dataset supporting the conclusions of this article is available from the corresponding author upon reasonable request.

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
