# Peer review of "Factors Related to Plasma Homocysteine Concentration in Young Adults: A Retrospective Study Based on Checkup Populations"

_jcm, 2023, doi:10.3390/jcm12041656_

Round 1

Reviewer 1 Report

My first review: The topic is actual, and adequately addressed through methods and criteria, and the manuscript is well written. The introduction section introduces the paper's subject, its research context, and its relevance. The discussion is well-balanced and includes relevant literature data. But the summary needs to be shortened and focused. Another request for the authors is to technically correct the tables.   Specific contents:

The manuscript deals with the factors (laboratory measurement and anthropometric assessment) related to plasma homocysteine concentration in young adults. This manuscript have protentional for the original point because this topic is not well understood. The introduction section introduces the paper's subject, its research context, and its relevance. The main topic is actual, and adequately addressed questionary through methods and criteria, the text is clear and easy to read, and the manuscript is well written. The discussion is well-balanced and includes relevant literature data. Authors concluded that males have significantly higher plasma Hcy level and HHcy prevalence than females in the young and data presented that age and LDL-C was negatively and BMI was positively associated with Hcy concentration in young males and that ALT, LDL-C, and Glu were negatively and AST, CREA, UA, and TG were positively correlated with plasma Hcy concentration in young females. The authors' conclusion is consistent with the presented evidence. But the summary needs to be shortened and focused. Another request for the authors is to technically correct the tables.

Reviewer 2 Report

This article investigates the distribution profile HHcy prevalence in males and females in subjects<40 years and demonstrates a higher prevalence in males rather than females as well as identifying a strong negative correlation with age which is noteworthy. This follows on from previous work by the group that identified males as an at risk group for HHcy

Overall, the paper is well written, methodology and statistical considerations appear sound and limitations are identified. However, extensive editing of English language is required.

Edits required:

Line 13 BMI Units are listed before the measure - this needs reversal

Lines 16-18 - it needs to be stated that these values are means and SDs (if this is the case).

Line 20-25 Strengths and statistical significance of relationships should be stated.

Line 31 et al is not correct terminology

Throughout manuscript where means are stated it should also be stated that SDs are also included

Line 101 - Do not use the word 'average' state that the value is 'mean +/- SD'

Line 103 - The term 'significant differences' is largely meaning less - state the direction and magnitude of difference.

Table 1- has a title for the penultimate column in Chinese characters - please change to English. 

Reviewer 3 Report

The study conducted by Zhihua Li et al. presents interesting results. The aim of the study was to assess the distribution profile and associations for plasma homocysteine level in the young adults.

During the reading of this article, I had major suggestions and questions:

1.       In the introduction, please write what is homocysteine and how is it formed.

2.       The obtained results indicate that the level of homecysteine ​​decreases with age. In turn, numerous literature data indicate that the level of homocysteine ​​increases with age. What could be the reason for the obtained results. What factors additionally affect the level of homocysteine in young people?

3.       Hyperhomocysteinemia may occur with a deficiency of B vitamins? Are there any data on vitamin B deficiency in the patients included in this study?

4.       Additionally, I propose to conduct a correlation analysis (Spearman or Pearson) between the level of homocysteine and each of the examined parameters (i.e. WHR, ALT, ASP etc.) for each age group separately.

5.       Table 1 should include the number of patients in each groups. In the table 1, data are presented as mean ± SD. Did the data possess a normal distribution? If the distribution of the data was not in accordance with the normal distribution, the data should be presented as the median (Quartile 1; Quartile 3)

6.       The description under Figure 1 is incomplete. There should be information: how the data are presented/expressed (average ± SD)? What statistical tests have been performed? If the distribution of the data was in accordance with the normal distribution, the data should be presented as Average ± SD with bar graphs with SD
